# Effect of Heating Time of Cottonseed Meal on Nutrient and Mineral Element Digestibility in Chicken (Based on Cottonseed Meal Replaced with All Soybean Meal)

**DOI:** 10.3390/ani12070883

**Published:** 2022-03-31

**Authors:** Xuean Xu, Haiming Yang, Zhi Yang, Zhiyue Wang

**Affiliations:** 1College of Animal Science and Technology, Yangzhou University, Yangzhou 225009, China; xuxuean1996@163.com (X.X.); dkwzy@263.net (Z.W.); 2Joint International Research Laboratory of Agriculture and Agri-Product Safety of Ministry of Education of China, Yangzhou University, Yangzhou 225009, China; zhiyang@yzu.edu.cn

**Keywords:** cottonseed meal, heat treatment, mineral element, nutrient digestibility

## Abstract

**Simple Summary:**

At present, we are faced with the problem of a feed protein resource shortage. Cottonseed meal is a potential alternative protein source for use in poultry diets. High-temperature treatment is a common processing method of cottonseed meal. Our experiment was conducted to estimate the effects of heating time of cottonseed meal on nutrient digestibility and mineral element absorption of chicken.

**Abstract:**

A digestibility test was conducted to estimate the effects of the heating time of cottonseed meal on nutrient digestibility and mineral element absorption in chicken. A total of 36, 45-week-old healthy New Yangzhou chickens with similar body weight were randomly divided into 6 groups with 6 replicates per group and one chicken per replicate. The chickens in Group A (control group) were fed the corn-soybean meal diet. The chickens in Groups B, C, D, E, and F (experimental groups) were fed the cottonseed meal to replace all soybean meal. The cottonseed meal in the experimental groups was treated with wet heating. The heating temperature was set at 120 °C, and the humidity was set at 50%. The heating time was set to 10, 15, 20, 25, and 30 min successively. The trial period was 4 day. The digestibility of crude protein, metabolic energy, and dry matter was highest using wet-heat treating for 15 min (*p <* 0.05). The digestibility of Fe increased significantly from 66.78% to 70.39% when the heating time of cottonseed meal was prolonged from 10 min to 30 min (*p <* 0.05). Compared with Group A, the digestibility of Zn was increased in Groups B and C, and then there was a decrease in Group D. Finally, the digestibility was increased again in Group F. The opposite pattern was shown in the digestibility of Cu and Zn. There was no significant effect of wet heat treatment time on the digestibility of calcium, phosphorus, and manganese (*p* > 0.05). When cottonseed meal is considered as feed for 45-week-old chickens, processing conditions of 120 °C and 50% humidity for 15 min are recommended.

## 1. Introduction

With the development of animal husbandry in China, poultry production has become more specialized and widespread. In recent years, animal husbandry has been faced with an increasing shortage of feed protein resources. The exploitation and application of new protein resources has attracted great attention in many countries [1]. As a good feed material, cottonseed meal is also a good-quality protein resource. But there are some harmful substances in cottonseed meal. If we can detoxify cottonseed meal effectively and use it reasonably, the problem of feed protein resource shortage can be solved to a large extent. Therefore, the detoxification technology of cottonseed meal and its application in livestock and poultry production need further study.

Cottonseed meal, an oil industry byproduct, is an alternative protein source for use in poultry diets [2]. The crude protein content of cottonseed meal is between 38% and 50%, the crude ash content is less than 9%, the crude fiber content is between 9% and 16%, and the crude fat content is less than 2.5% [3]. Furthermore, cottonseed meal contains a variety of amino acids. The content of lysine is about 1.6% less than that of soybean meal by 2.1%, and the content of methionine is about 0.5%. The contents of arginine, proline, and phenylalanine are higher than those of soybean meal. In addition, cottonseed meals also contain a variety of mineral elements and vitamins [4].

Cottonseed meal is widely used as a feed raw material. A study found that using cottonseed meal instead of soybean meal to feed tilapia did not affect growth performance [5]. Cottonseed meal is also commonly used instead of soybean meal in poultry farming. It was reported that 6.73 to 26.91% cottonseed meal in feed improved the growth performance of goslings from 28 to 70 day [6]. Cottonseed meal can replace up to 75% of soybean meal without adverse effects on performance, hematological values, and carcass quality of 8-week-old broilers [7].

However, cottonseed meal contains a variety of anti-nutritional factors, which may cause animal health problems. Batonon used cottonseed meal instead of soybean meal to feed broilers. He found that when the contemporary replacement was 40%, the feed intake of chickens increased, but the growth performance decreased [8]. Free gossypol (FG) is the most important anti-nutritional factor in cottonseed meal [9]. Free gossypol, as a polyphenolic compound, reduces protein digestibility by inhibiting pepsinogen, pepsin, and trypsin activity in the gastrointestinal tract, and binds iron in the diet [2,9]. Zeng found that when the content of free gossypol in the diet was too high, the reproductive performance of poultry was impaired [10]. The difference in tolerance to cottonseed meal in poultry may be related to broiler species, age, and source of cottonseed meal.

At present, there are three main methods to eliminate free gossypol: physical, chemical, and biotechnology methods. Wet heating is a traditional physical method. Hu studied the effect of extrusion temperature (90, 100, 110, 120, 130 °C) on free gossypol in cottonseed meal. He found that an extrusion temperature of 120 °C could not only significantly reduce the content of free gossypol but also be conducive to maintaining the nutritional value of cottonseed meal [11]. When the temperature exceeded 120 °C, the degradation rate of free gossypol no longer increased with an increase in temperature. Studies found that the detoxification rate of free gossypol in cottonseed meal was increased linearly accompanied by a temperature rise at a constant humidity, and there was a significant interaction between the temperature and humidity [12]. Several reports have identified a better dephenolization effect on cottonseed meal at 120 °C, in 50% humidity when heating for 30 min [13]. New Yangzhou chicken is a new breed of sanhuang chicken bred by the Jiangsu Agricultural College for 23 years on the basis of yangzhou local chicken breeds, through breeding, crossbreeding, strain breeding, and so on. It has the advantages of fresh meat quality, fast growth rate, and strong vitality. The aim of this study was to investigate the effects of heating time of cottonseed meal on nutrient digestibility and mineral element absorption of New Yangzhou male chickens, providing a scientific basis for efficient and reasonable digestibility of cottonseed meal and direction for solving the problem of shortage of high-quality protein resources in animal feed.

## 2. Materials and Methods

### 2.1. Ethics Statement

All bird-handling protocols were approved by the Yangzhou University Animal Experiment Ethics Committee, with permit number SYXK (Su) IACUC 2021-0029. All bird experimental procedures performed by the Regulations for the Administration of Affairs Concerning Experimental Animals were approved by the State Council of the People’s Republic of China.

### 2.2. Experimental Design and Diets

A total of 36 healthy male *New Yangzhou* chickens of 45 weeks old with similar weight were randomized to six groups with six replicates per treatment and one bird per replicate. Group A was the control group and was fed the basal diet. Groups B, C, D, E, and F were the experimental groups. The cottonseed meal in their diet was treated by heating. The heating temperature was set at 120 °C, and the humidity was set at 50%. The heating time was set to 10, 15, 20, 25, and 30 min, successively. The pre-feeding period was 4 day. The trial period was 4 day. Artificial feeding and free feeding. The chickens were indoor reared under suitable and similar environmental conditions. Birds had free access to food and water for 24 h under light (12 h–dark (12 h) conditions. The room temperature was approximately 20 °C, and no heat was provided. The total feces collection method was adopted in this test [14]. The diet was based on the NRC (1994) [15]. The composition and nutrient levels of the experimental diets are listed in Table 1.

### 2.3. Sample Collection and Analytical Processing

The total excreta collection method was used. Each chicken was placed in a separate cage. A tray was placed under each cage to collect feces. The feed intake was recorded on a daily basis. The feces were collected at 8 am every day. The feathers, dandruff, and feed were removed from the plate when collecting feces. Hydrochloric acid (10%; 10 mL/100 g) was added to the plastic bags containing feces. The feces in each group were collected for four days, and then the samples were dried in the oven at 65 °C, and the moisture was returned for 24 h under the natural state. After being crushed, the samples were made into air-dried samples. The initial feed sample was stored at −20 °C for laboratory analysis. Excreta samples were frozen until required for the chemical analysis described by Zhang [16].

All samples were analyzed for crude protein (CP), apparent metabolizable energy (AME), ether extract (EE), dry matter (DM), calcium (Ca), total phosphorus (TP), and partial trace elements, including Fe, Zn, Mn, and Cu. Crude protein content was calculated based on the nitrogen content, according to the method of Kjeldhal [17]. Gross energy content was determined by an oxygen bomb calorimeter (IKA-C2000, Staufen, Germany). Dry matter content was determined after drying at 100 °C for 48 h. The content of total phosphorus in the samples was determined by ammonium molybdate spectrophotometry with nitric acid-perchloric acid as an oxidant. Calcium content in the samples was determined by EDTA titration [17]. About 0.5 g of each feces sample was weighed in triplicate and digested with 10 mL HNO_3_ and 0.4 mL HClO_4_ at 200 °C in a 50-mL calibrated flask until the solutions cleared. They were then dried at 100 °C for 24 h and dry-ashed for 10 h at 550 °C. The ashed samples were dissolved in a nitric acid–perchloric acid mixture (1:1) and diluted with ddH_2_O for mineral analysis. The content of Cu, Fe, Mn, and Zn was measured using flame atomic absorption spectrophotometry (Analytik JenanovAA 400P, Jena, Germany). The sample was digested at a low temperature for 1 h and cooled. One milliliter perchloric acid was added to the sample and heated until the solution was almost transparent, then cooled again. Lanthanum nitrate solution was added and transferred to a 50 mL volumetric flask, and the concentration of 0.15 mol/L nitric acid medium, and the mass fraction of 0.22% lanthanum nitrate releasing agent medium were maintained. Direct determination was made by a flame atomic absorption spectrophotometer. Five copies of each sample were made in parallel, and the average value was taken.
Coefficient of retention = (Feed intake × Nutrient_diet_ − Excreta output × Nutrient_excreta_)/(Feed intake × Nutrient_diet_) × 100%

### 2.4. Data Analysis and Processing

All data were analyzed using a one-way ANOVA in SPSS 17.0 [18]. Each replicate pen served as an experimental unit for all statistical analyses. Data were subjected to one-way analysis to determine linear and quadratic responses to increasing the time of wet heat treatment of cottonseed meal. Significant differences among the treatment means were determined at *p* < 0.05 by Tukey’s multiple range tests.

## 3. Results

### 3.1. Digestibility of CP, AME, and DM

The effects of time of wet heat treatment of cottonseed meal on the digestibility of CP, AME, and DM are shown in Table 2. The heating time of cottonseed meal had significant effects on the digestibility of CP, AME, and DM (*p* < 0.05). The digestibility of CP, AME, and DM had a quadratic relationship with heating time. Compared to the control group, there was a markedly digestibility elevation of CP, AME, and DM when heated for 15 min. The digestibility of CP in Group C (70.27%) was 20.95% and 25.70% higher than that in Group A and Group F (*p* < 0.05). The digestibility of AME in Group C was higher than that in Group A (*p* < 0.05). The digestibility of DM in Group C (79.85%) was 30.34% higher than that in Group A (*p* < 0.01).

### 3.2. Digestibility of Ca and TP

The effects of the time of wet heat treatment of cottonseed meal on the digestibility of Ca and TP are shown in Table 3. The effects of heating time of cottonseed meal on the digestibility of Ca and TP were not significant (*p* > 0.05).

### 3.3. Digestibility of Fe, Cu, Mn, and Zn

The effects of time of wet heat treatment of cottonseed meal on the digestibility of Fe, Cu, Mn, and Zn are shown in Table 4. The effects of the heating time of cottonseed meal on the digestibility of Mn were not significant (*p* > 0.05). The digestibility of Fe increased significantly with a longer heating time of cottonseed meal (*p* < 0.05). The digestibility of Fe in Group F was the highest (70.39%). Compared with Group A, the digestibility of Zn increased in Groups B and C, and then there was a decrease in Group D. Finally, the digestibility was increased again in Group F. The digestibility of Zn was highest (33.44%) when the time of high-temperature treatment of cottonseed meal was 30 min (*p* < 0.01). In contrast, the digestibility of Cu decreased first, and then there was a rise between Groups C and D. Then, the digestibility of Cu decreased again.

## 4. Discussion

Numerous studies have shown that cottonseed meal has no adverse effects on performance, hematological values, and carcass quality of the chicken [7], but cottonseed meal contains a variety of anti-nutritional factors, such as free gossypol, which may affect its nutritional value. Other methods, such as pelleting, extrusion, cooking, and Ca(OH) treatment of cottonseed meal, might improve feeding value in poultry [19]. In this study, soybean meal was completely replaced by cottonseed meal. The current study aimed to investigate the effects of the heating time of cottonseed meal on the nutrient digestibility and mineral element absorption of chicken, including CP, AME, DM, Ca, TP, Fe, Cu, Mn, and Zn. Our study found that wet heat treatment of cottonseed meal can affect its nutritional value.

At present, damp-heat detoxification is a common processing method of cottonseed meal, which has the advantages of simplicity and low cost. The activity of antinutritional factors in cottonseed meal was weakened when cottonseed meal was heated. This could lead to a reduction in the inhibitory effect on the use of nutrients. Our results indicate that wet heat treatment of cottonseed meal for 15 min significantly improved the digestibility of crude protein. Gossypol can inhibit the activities of pepsin and trypsin in the gastrointestinal tract, thereby inhibiting their activities and reducing the digestibility of protein [19]. Free gossypol can inhibit the activity of various proteases and bind to proteins to reduce the digestibility of proteins. Heating cottonseed meal for 15 min can effectively reduce the content of free gossypol. This indicates that wet heat treatment of cottonseed meal for 15 min can significantly improve the digestibility of metabolizable energy. Previous studies have found that the performance of broilers could be consistent with that of soybean meal by increasing protein content in the cottonseed meal diet. These results indicated that the addition of cottonseed meal to the broiler diet may cause a decrease in dietary protein availability. Increasing protein content can compensate for this loss. In particular, lysine deficiency can directly affect the growth performance of broilers. This will affect the digestibility of feed energy. We speculate that the content of free gossypol affects the taste of feed and the intake of animals. The highest dry matter digestibility was shown in Group C. The digestibility of DM depends mainly on the digestibility of crude protein and energy. Therefore, wet heat treatment of cottonseed meal for 15 min could significantly improve the digestibility of dry matter. However, the digestibility of CP, AME, and DM descend when the heating time exceeded 15 min. There was an overly low content of free gossypol in the cottonseed meal when heating for 15 min, but when the cottonseed meal was heated, it led to the decline of protein quality, especially lysine [20]. Fernandez et al. demonstrated that overheating can reduce amino acid digestibility of cottonseed meal and affect protein quality [21]. Originally, the content of lysine and methionine in cottonseed meal was low. Therefore, the effect of high temperature on anti-nutritional factor was unobvious when the cottonseed meal was heated for over 15 min. Instead, high temperature can destroy the protein composition of cottonseed meal, affect the taste of feed, and reduce feed intake of animals, which leads to the reduction of the digestibility of CP, AME, and DM.

Dietary mineral elements play an important role in poultry production. Improper application of dietary mineral elements not only leads to the decline of poultry performance but also causes resource waste and environmental pollution. Calcium and phosphorus are the most abundant mineral elements in livestock and poultry, and they have complex interactions in digestion, absorption, metabolism, digestibility, and excretion. There are various minerals in the cottonseed meal, including low-content Ca and high-content TP. The findings showed that the apparent digestibility of calcium and phosphorus was significantly improved in the yellow-feather broiler feed in the cottonseed meal [22]. However, the effects of heating time on the digestibility of calcium and phosphorus were not significant in this test. At present, the mechanism of the effect of gossypol on the digestibility of calcium and phosphorus in feed is unclear. However, other anti-nutritional factors contained in cottonseed meal, such as phytic acid, can affect the digestibility of calcium and phosphorus [23]. Although the digestibility of calcium and phosphorus in Group C were the highest in this experiment, there was no significant difference between Group C and the other groups. Further discussion and research are needed.

Cottonseed meal samples were analyzed, and the mean proximate composition of minor minerals (mg/kg) are: copper (0.03), iron (118.50), manganese (58.22), and zinc (62.25) [24]. The mechanism of coordination and antagonism among Cu, Fe, Zn, and Mn has been fully analyzed, and its essence is that one element affects the absorption and digestibility of other elements. The wet heat treatment has a certain effect on the digestibility of mineral matter. Many antinutritional factors in cottonseed meal can affect the digestion and absorption of minerals. Free gossypol can bind to protein and iron, causing some enzymes to lose activity, while binding with iron can interfere with hemoglobin synthesis and cause iron deficiency anemia [25]. Studies have shown that adding ferrous sulfate to cottonseed meal diets can effectively improve the growth performance of broilers [26]. Braham reported that gossypol can cause anemia in pigs, and proper supplementation of calcium and iron ions can eliminate the symptoms of anemia, which also indicates that gossypol can bind to metal ions [27]. In this study, the digestibility of Fe increased significantly with a longer heating time of cottonseed meal. There are many factors affecting the digestibility of Zn, such as the type of food, the protein level, and the additive amount. In our experiment, the digestibility of Zn was increased, and then there was a decrease between Groups C and D; finally, the digestibility increased again. The opposite pattern was shown in the digestibility of Cu and Zn. Studies have found that excessive Cu concentration can inhibit the absorption and digestibility of Zn and vice versa [28]. The results indicate that these two elements have competitive and inhibitory effects on each other. Mineral elements, including Cu, Fe, and Mn, play an important role in poultry production. The essential minerals must be provided in the diet at concentrations sufficient optimum response, and the dietary intake of other minerals must be low enough to provide complete safety for both the animals and the human population [29,30]. Therefore, the digestibility of dietary mineral elements in poultry is worthy of further exploration to develop adequate feed digestibility efficiency.

## 5. Conclusions

In summary, the aim of this study was to provide a reference for the application of cottonseed meal in the diet of chickens. The wet heat treatment of cottonseed meal under different conditions had an effect on its nutritional absorption. When cottonseed meal is considered feed ingredient to replace soybean meal for adult chickens, processing conditions at 120 °C and 50% humidity for 15 min are recommended.

## Figures and Tables

**Table 1 animals-12-00883-t001:** Composition and nutrient levels of the experimental diets (air-dry basis).

Ingredients	Value (%)	Nutritional Levels	Value (%)
Corn	62.40	Metabolizable Energy, (MJ/Kg)	11.68
Cottonseed meal (46%)	25.00	Crude protein	16.80
Wheat bran	3.50	Calcium	1.54
CaHPO_4_	1.30	Total phosphorus	0.77
Stone powder Limestone	3.00	Available phosphorus	0.47
*DL*-methionine	0.16	Lysine	0.81
Salt	0.30	Methionine	0.35
Soya-bean oil	3.00		
*L*-lysine hydrochloride	0.34		
Premix ^†^	1.00		
Total	100		

^†^ The premix provided the following per kg of diets: VA 8000 IU, VD_3_ 1500 IU, VE 30 IU, VK_3_3 mg, VB_1_4 mg, VB_2_6 mg, VB_6_4 mg, VB_12_30 µg, nicotinamide 35 mg, calcium pantothenate 15 mg, folic acid 2 mg, Fe 60 mg, Mn 80 mg, Zn 80 mg, I 0.4 mg, Cu 10 mg, Se 0.3 mg.

**Table 2 animals-12-00883-t002:** Effects of heating time of cottonseed meal on the digestibility of Crude protein(CP), Apparent metabolizable energy(AME), and Dry matter(DM).

Item	Group						SEM	*p*-Value		
	Group A	Group B	Group C	Group D	Group E	Group F		Groups	Linear	Quadratic
AME (MJ/kg)	10.62 ^b^	10.74 ^ab^	11.24 ^a^	11.12 ^ab^	10.66 ^ab^	10.41 ^ab^	0.272	0.021	0.088	0.015
CP (%)	47.10 ^ab^	51.00 ^ab^	56.97 ^a^	50.94 ^ab^	49.67 ^ab^	45.32 ^b^	1.265	0.043	0.122	<0.001
DM (%)	61.26 ^c^	66.74 ^b^	79.85 ^a^	76.16 ^a^	69.39 ^b^	67.99 ^b^	1.313	0.007	0.024	0.008

Chickens in group A were fed the corn–soybean meal diet. Chickens in groups B, C, D, E, and F were fed the cottonseed meal to replace all soybean meal, and represent cottonseed meal wet heat treatment times of 0, 15, 20, 25, and 30 min, respectively. SEM is the standard error of the mean. The results are average values; the same letter or no letter in the same column indicates that the difference is not significant (*p* > 0.05), and different lowercase letters indicate a significant difference (*p* < 0.05).

**Table 3 animals-12-00883-t003:** Effects of heating time of cottonseed meal on the digestibility of Ca and P %.

Item	Group						SEM	*p*-Value		
	Group A	Group B	Group C	Group D	Group E	Group F		Groups	Linear	Quadratic
Ca	58.74	60.77	62.01	60.70	60.59	60.15	1.266	0.087	0.152	0.136
TP	47.52	50.33	51.19	48.00	43.18	45.90	0.303	0.145	0.174	0.056

Chickens in group A were fed the corn–soybean meal diet. Chicken in groups B, C, D, E, and F were fed the cottonseed meal to replace all soybean meal, and represent cottonseed meal wet heat treatment times of 0, 15, 20, 25, and 30 min, respectively. SEM is the standard error of the mean. The results are average values; no letter in the same column indicates that the difference is not significant (*p* > 0.05).

**Table 4 animals-12-00883-t004:** Effects of heating time of cottonseed meal on the digestibility of trace elements %.

Item	Group						SEM	*p*-Value		
	Group A	Group B	Group C	Group D	Group E	Group F		Groups	Linear	Quadratic
Fe	65.43 ^b^	66.78 ^b^	67.45 ^b^	67.56 ^b^	70.29 ^a^	70.39 ^a^	5.492	0.042	0.011	0.218
Zn	25.24 ^bc^	25.51 ^ab^	27.85 ^ab^	21.92 ^c^	26.42 ^ab^	33.44 ^a^	0.950	0.016	0.025	0.071
Mn	18.87	20.50	28.85	24.89	17.65	22.17	1.145	0.230	0.278	0.451
Cu	74.98 ^ab^	75.19 ^a^	53.53 ^c^	56.64 ^c^	63.38 ^b^	44.29 ^c^	2.736	<0.001	<0.001	0.124

Chickens in group A were fed the corn–soybean meal diet. Chickens in groups B, C, D, E, and F were fed the cottonseed meal to replace all soybean meal, and represent cottonseed meal wet heat treatment times of 0, 15, 20, 25, and 30 min, respectively. SEM is the standard error of the mean. The results are average values; the same letter or no letter in the same column indicates that the difference is not significant (*p* > 0.05), and different lowercase letters indicate a significant difference (*p* < 0.05).

## Data Availability

Data are available on reasonable request from the corresponding author.

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
