# Peer review of "Effect of Heating Time of Cottonseed Meal on Nutrient and Mineral Element Digestibility in Chicken (Based on Cottonseed Meal Replaced with All Soybean Meal)"

_animals, 2022, doi:10.3390/ani12070883_

Round 1

Reviewer 1 Report

The article “Effect of Heating Time of Cottonseed Meal on Nutrient and Mineral Elements Utilization in Chicken (based on cottonseed meal replaced all soybean meal)” by Xu et al needs significant improvements. The introduction needs to concise and reorganised in a concise manner. Simple summary and conclusions should also be improved. Besides, there are additional comments that must be addressed by the authors.

Line 19: Please check font size of words “metabolism test”.

Lines 20-22: Please check grammatical structure of the sentence.

Line 27: Please remove the word “and” between the words “of” and “iron”.

Lines 32-34: Please improve the conclusive statements in the abstract.

Line 39 & 58-59: Please check the grammatical structure of the sentence.

Line 70: The word “damaged” is not suitable. Please replace with a suitable word.

Line 76: Please clarify whether high temperature reduces or maintains the nutritional value of cottonseed meal.

Lines 81-83: Please check for grammatical errors.

Lines 84-86: Please state the importance and objectives of the study instead of repetitive statements.

Lines 89-92: Please remove these lines.

Line 93: Please correct the word “handing” as “handling”.

Lines 101-104: These lines should be part of introduction. Please remove from here.

Lines 111-112: Please clarify whether the light was provided for 24 hours? Also mention the duration of darkness if provided.

Lines 123-126: Please check grammatical structure of the sentences.

Line 131: Please clarify what is meant by "obtain nitrogen".

Lines 136-137: Please clarify whether feces samples collected were 0.5 g of 0.4 g?

Line 172: Please insert the word "of" between the words "effect" and "heating".

Lines 195-201: The statements are repeating. These statements are also mentioned in the introduction. Please remove to avoid duplication.

Lines 184-291: Discussion part needs to be improved significantly in the wake of the results of the study and previous scientific reports.

Lines 283-286: Conclusions are repeating statements that must be improved.

Author Response

Dear reviewer

Thank you very much for your comments on our manuscript, and we are very grateful to your constructive comments. The detailed responses is attached.

Thank you very much for your time and consideration.

Yours Sincerely,

Xuean Xu & Haiming Yang

Reviewer 2 Report

It is good piece of reaserch with clear aims and objectives but methodology is poorly explained. The following information is missing:

 Line 91 -92: Give reference to the EU standards used.

Line 99: why 45 old chickens were used?

Line 99: Were chickens males or females?

Line102: Is Yangzhou local chicken breeds a dual breed?

Line 104: What do you mean by “strong living force”? Not clear.

Line 108: Why pre-feeding and trial periods were 4 d each? Do you have a reference for the procedure used?

Line 109: what is the conventional method of feeding?

Line 110: These chickens were indoor rearing? Where were these chickens until day 45? Were they raised in a cage or floor pens? What were the pen dimensions? And what was cage dimension when they were moved to a cage?

Line 124: Put feces in plastic bags and add 10% hydrochloric. Why was HCL added? The reason should be explained and give a reference.

Line 129: what do you mean by energy expenditure (EE)? Not clear?

Line 130: You have mentioned that samples were analysed for trace elements. This is not true. You have not analysed all trace minerals. Be specific and say precisely that Fe, Zn, Mn and Cu were analysed.

Line 133: Dry matter content was determined after drying at 65℃? Why? DM is typically determined at 100℃. Why was lower temperature used? Give reference?

Line 126-127: Why air-dried samples were frozen? By freezing the samples, moisture will be reintroduced in them? Not clear what was done? Clarify?

Line 160: What is ME? And how was it determined? There is no mention of ME in the methods and material section. How was ME determined?

Line 171: This is the first time the wet heat treatment is mentioned? If it was a wet heat treatment, explain how this wet treatment was applied?  

Line 259: cu should be with capital C.

271: Such “us” I think you such as?

Line 282: Conclusion section to be re-written. Not all nutrients were improved, so be specific

Author Response

(The authors gave the same response as above.)

Reviewer 3 Report

The manuscript has merit for dealing with a subject of great interest today, which is the search for new sources of dietary proteins. I just missed a greater detail on the cottonseed meal used, its composition and method of obtaining it. Detail the procedure of the heat treatments studied. Another point would be to better describe the animal, make it clear that they were males.

Author Response

Dear reviewer

Thank you very much for your comments on our manuscript, and we are very grateful to your constructive comments. We added the introduction of heat treatment of cottonseed meal and emphasized that the animal was male.

Thank you very much for your time and consideration.

Yours Sincerely,

Xuean Xu & Haiming Yang

Reviewer 4 Report

The authors have conducted a study on the use of cottonseed meal in chickens. Please consider the following comments:

Comment 1: The data must be analyzed to study the trends and the influence the increasing time has on the results (regressions; trends; optimum responses), not only to understand the differences among specific heating times. The usefulness of knowing specific treatment differences is minimal. Please reanalyze your data and redesign the tables with results.

Comment 2: Please update the whole manuscript considering the previous comment.

Comment 3: The study is a digestibility one, not a metabolism study or test. Please remove statements regarding metabolism where it is not appropriate, for example, at the beginning of the abstract.

Comment 4: Please do not confuse digestible with metabolizable. Check the whole manuscript carefully.

Comment 5: The titles of tables 2, 3, and 4 say “on the utilization of”. Please remove any statement about “utilization” from your study: you have measured digestibility, not utilization.

Author Response

(The authors gave the same response as above.)

Round 2

Reviewer 1 Report

The article “Effect of Heating Time of Cottonseed Meal on Nutrient and Mineral Elements Utilization in Chicken (based on cottonseed meal replaced all soybean meal)” by Xu et al needs improvements. Methodology needs to be improved significantly. In addition, authors must address the following comments:

Line 22 & 23: “The chicken in groups B, C, D, E, and F (experimental groups) were treated with wet heating.” Whether the chickens were treated with wet heating? If yes, why?

Lines 31-32: “Heat treatment of cottonseed meal can affect its nutritional value.” Please mention specific effects.

Line 67-69: “Zeng found when the content of free gossypol in diet is too high, the reproductive performance of poultry will be impaired and lost [10].” What is meant by the word "lost" in this statement. Please explain.

Lines 80-82: “Several reports have identified there was a better dephenolization effect on cottonseed meal is better at 120℃, in 50% humidity and heating 30min [13].” Please check for grammar.

Line 83-86: Please state the importance and objectives of the study. Also explain the context of "breeding, crossbreeding, strain breeding and so on".

Line 104-106: “Feed these chickens according to the conventional method. These chickens were indoor rearing under suitable and similar environmental conditions. Birds had free access to diets and water for 24 h under light (12hrs)-dark (12hrs) conditions. The birds were maintained under natural daylight.” Line 104 states that birds were reared indoor. Line 106 states birds were maintained under natural light.” Please give details how the natural light was provided to the birds reared indoor and provide a reference/standard protocol or approved code for the duration of light.

Line 118 & 119: “10% hydrochloric acid (10ml/100g) was added in plastic bags containing feces.” Why hydrochloric acid was added to the feces?

Lines 127-128: Dry matter content was determined after drying at 100℃for 48h. What was the percentage of dry matter?

Lines 84-86: “About 0.5 g of each feed and feces sample and 0.5 g of each feces sample were weighed in triplicate…….”. The words "and feces sample" are written twice in the same statement. Please remove the duplicated words.

Lines 273-275: Conclusions sections must be improved and future perspectives of the study should be mentioned.

Author Response

(The authors gave the same response as above.)

Reviewer 2 Report

Dear authors,

The manuscript is considerably revised, but your abstract and methods and material section require substantial improvement. 

Abstract: Need to be rewritten.

Line 22: “consider rewriting, “The chicken in groups B, C, D, E, and F (experimental groups) were treated with wet heating.” This is not correct. I think you meant feed was given a wet heat treatment? The chicken will die if you provide them with heat treatment at 120C.

Add a line to say how long the experimental diets were fed.

Line 27: “increased significantly with longer wet-heat treating” clarify how many minutes? The term “longer” is unclear.

Line 27: what do you mean by “Initially the digestibility rate of zinc was increased”? Digestibility rates were only analysed once at the end of the study, so what does initially means here? Consider revising the sentence.

Line 30: Revise sentence. “ No not significant”?

Line 32: considered as chicken feed: Please be specific and mention the age of chicken as the digestibility rate of heat-treated CSM may be different at different age birds.

Line 58-60: Please clarify at what age and in which species?

Line 78-79: “increase linearly accompanied by temperature riseat the same humidity” What do you mean by temperature rise at the same humidity? I think you mean at constant humidity? Also, give space between the rise and at.

Line 85: As male chickens were used in your study, therefore, do not repeat referring to high egg-laying performance,

Line 88: mention “male” chicken. So that reader is clear that this study was conducted using cockerels.

Line 102: which feed was given to birds during 4 d pre-feeding period? Not clear

Line 103: what are conventional methods? Clarify.

Line 117: The method should be written in passive voice (third person).

Line 120: How samples were air-dried? Not clear.

Line 145: also, add the formula for how ME was determined. I think by ME, you mean AME?

Line 193: birds? Not clear which species and at what age?

In the discussion section, when you are reporting literature, it would help if you would mention which specific birds (broilers, layers, breeders) and their ages. All you need to do is read the referenced paper and provide the additional information.

Line 274: In conclusion mention the name of the breed and the age.

Author Response

(The authors gave the same response as above.)

Reviewer 4 Report

Please consider the following comments:

Comment 1: The title should say "on nutrient and mineral elements digestibility in chicken..." as digestibility, not utilization, was evaluated. 

Comment 2: Every single mention of utilization in the manuscript that comes from your study must be removed. Use digestibility instead. The reason is you have not tested utilization. Please carefully check the whole manuscript and ensure to satisfy this observation as this has been pointed out in a previous review. This is a major flaw. If failing to correct this issue in the whole manuscript, the authors risk the manuscript being rejected.

Comment 3: Please add to 2.4 Data analysis and processing, what corresponds to the analysis of linear and quadratic trends.

Comment 4: Please ensure past tense is used in the whole manuscript regarding results of your study, like "cottonseed meal increased linearly" in line 78.

Comment 5: Please add footnotes to each of the tables to clarify the definition of each of the tested groups.

Comment 6: Line 238 should say "The findings showed that the apparent digestibility of calcium and phosphorus..."

Comment 7: Line 25 should say "The digestibility of crude protein, energy, and dry matter was..."

Comment 8: Rather than "digestibility rate" please use "digestibility". Adding "rate" makes it redundant. Please carefully check the whole manuscript.

Author Response

(The authors gave the same response as above.)

Round 3

Reviewer 1 Report

The article requires significant improvement in various sections. I have following comments that must be addressed by the authors.

As already suggested in previous comments, please give a mention of the effects of heat treatment on nutritional quality of cottonseed meal.

Line 85: What is the context of the words "breeding, crossbreeding, strain breeding and so on" in this study? Please remove these words.

Line 86-90: Aims of the study are not clear. Please state the importance and objectives of the study clearly.

Line 107-108: “Birds had free access to diets and water for 24 h under light (12hrs)-dark (12hrs) conditions.” I do not agree with duration of light and dark period for growth of chickens? Please provide a reference/standard protocol or approved code for the duration of light.

Lines 287-291: What do the conclusions sections achieve in the context of this study? As suggested in previous comments, conclusions sections must be improved and future perspective of the study should be mentioned.

Reviewer 2 Report

Dear Authors, 

The technical component of the manuscript is improved but requires spell and grammar check. I strongly suggest going through the whole manuscript and improving the English language. Few areas were corrections are required are as bellow:

Line 23: “were fed the cottonseed meal replace all soybean meal. “ something is missing here? Revise sentence. Suggest adding to before replace?

Line 88-89: New Yangzhou male chicken, which provide scientific basis for efficient and reasonable digestibility of cottonseed meal, and provide direction for solving 89 the problem of shortage of high-quality protein resources in animal feed.: Change the wording and suggest using plural verb form “provides.” In both places.

Line 106: Artificial feeding and free feeding.? Incomplete sentence. I think the authors meant to say that the birds had adlib access to feed and water?

Line 122:  The samples were dried in the oven at 65℃, and the moisture was returned for 24 hours under the natural state. Still not clear how moisture was returned? What is a natural state?

Line 168  & 198: fed the cottonseed meal replace all soybean meal? Revise sentence

Line257:  Although the digestibilitys of calcium and phosphorus. Spell check. The word is digestibilities

Conclusion:  When cottonseed meal is considered as chicken feed? Consider revising the sentence. Cottonseed is not a feed, it’s a feed ingredient or a protein source that, after wet heat treatment, can replace soybean meal for adult chickens.

Reviewer 4 Report

Well done.

This manuscript is a resubmission of an earlier submission. The following is a list of the peer review reports and author responses from that submission.